# Physiological responses of *Skeletonema costatum* to the interactions of seawater acidification and combination of photoperiod and temperature

Hangxiao Li[1,2], Tianpeng Xu[1,2], Jing Ma[1,2], Futian Li[1,2,3]* & Juntian Xu[1,2,4]*

[1]Jiangsu Key Laboratory of Marine Biotechnology, Jiangsu Ocean University, Lianyungang 222005, China
[2]Jiangsu Key Laboratory of Marine Bioresources and Environment, Jiangsu Ocean University, Lianyungang 222005, China
[3]Jiangsu Institute of Marine Resources, Lianyungang 222005, China
[4]Co-Innovation Center of Jiangsu Marine Bio-industry Technology, Jiangsu Ocean University, Lianyungang 222005, China

*Correspondence to*: Futian Li (futianli@jou.edu.cn) and Juntian Xu (jtxu@jou.edu.cn)

**Abstract.** Ocean acidification (OA), which is a major environmental change caused by increasing atmospheric $CO_2$, has considerable influences on marine phytoplankton. But few studies have investigated interactions of OA and seasonal changes in temperature and photoperiod on marine diatoms. In the present study, a marine diatom *Skeletonema costatum* was cultured under two different $CO_2$ levels (LC, 400 µatm; HC, 1000 µatm) and three different combinations of temperature and photoperiod length (8:16 L:D with 5 °C, 12:12 L:D with 15 °C, 16:8 L:D with 25 °C), simulating different seasons in typical temperate oceans, to investigate the combined effects of these factors. The results showed that specific growth rate of *S. costatum* increased with increasing temperature and daylength. However, OA showed contrasting effects on growth and photosynthesis under different combinations of temperature and daylength: while positive effects of OA were observed under spring and autumn conditions, it significantly decreased growth (11 %) and photosynthesis (21 %) in winter. In addition, OA alleviated the negative effect of low temperature and short daylength on RbcL and key PSII proteins (D1 and D2). These data indicated that future ocean acidification may show differential effects on diatoms in different cluster of other factors.

**Key words.** diatom, growth, photosynthesis, $CO_2$, temperature and photoperiod

## 1 Introduction

Ocean acidification (OA) is one of major environmental changes caused by increasing atmospheric $CO_2$, which has directly raised from 280 ppm in preindustrial era to higher than 400 ppm at present (Friedlingstein et al., 2019). It is predicted that surface seawater pH would drop 0.3–0.5 and 0.5–0.7 units by the year 2100 and 2300 respectively (Caldeira and Wickett, 2003). It has been suggested that calcifying organisms, such as coral reefs and coccolithophores, are vulnerable to OA due to the decreased calcification at elevated $CO_2$ (Albright et al., 2016). The responses of non-calcifying organisms such as

diatoms to OA vary widely among taxonomic groups which may be detrimental, negligible or even beneficial (Gao and

Campbell, 2014). Consequently, the abundance of marine phytoplankton and community structure might be altered by OA (Gattuso et al., 2015).

Diatoms are ubiquitous photosynthetic phytoplankton which account for about 20 % of global primary productivity, and thus play a crucial role in the global cycling of carbon and silicon (Falkowski et al., 2004). To overcome the limited aqueous $CO_2$ concentration in seawater, they have developed $CO_2$-concentrating mechanisms (CCMs) (Spalding, 2007). Decreased

photosynthetic affinity for dissolved inorganic carbon (DIC) and activity of CCM related enzymes are generally found under increased $CO_2$ condition (Raven and Beardall, 2014). For phytoplankton assemblages, elevated $CO_2$ could lead to increases in chlorophyll *a* concentrations and the abundance of diatoms (Johnson et al., 2013). Species and strain specificity are observed in studies on physiological responses of diatoms to OA, which might be caused by the balance between positive effects of elevated $CO_2$ and negative effects of decreased pH (Langer et al., 2009; Li et al., 2016). In addition, acclimation

and adaptation processes, i.e. the timescale of diatoms exposed to OA, could also influence the physiological effects of OA (Wu et al., 2014; Li et al., 2017). Moreover, other environmental factors, such as temperature (Seebah et al., 2014), light (Gao et al., 2012), nutrients (Li et al., 2015) and clusters of multiple factors (Xu et al., 2014a; Xu et al., 2014b), are shown to have interaction with OA on diatoms.

Diatoms are widespread across oceans, thus they would experience different photoperiods. Photoperiod controls the total

light dose received by phytoplankton and thus could remarkably influence the physiological performance such as growth and lipid content of microalgae (Wahidin et al., 2013). For Antarctic sea ice microalgae *Chlamydomonas* sp., continuous illumination stimulates higher growth and nutrient absorption rates than successive darkness condition (Xu et al., 2014c). Growth rate of *Chlamydomonas reinhardtii* is gradually enhanced following the increasing photoperiod (Hsieh et al., 2018). In contrast, *Alexnadrium minutum* grows faster under short daylength relative to longer and even continuous daylength

(Wang et al., 2019). Moreover, different photoperiods could influence intracellular carbon demand of microalgae, which has a stronger regulation effect on CCMs compared with effects of changes in $CO_2$ supply (Rost et al., 2006).

Under the combined influence of photoperiod and OA, physiological performance of phytoplankton might be different from that under single factor. For example, continuous light moderates the negative effect of OA on coccolithophore growth, although species isolated from different regions show diverse responses (Bretherton et al., 2019). The changes of

photoperiod are often accompanied by increase or decrease temperature, and impacts of OA on diatoms can also be changed by temperature. For example, under OA condition, decreased metabolic activity is observed in *Phaeodactylum tricornutum* when temperature elevates (Bautista et al., 2018), while elevated temperature enhances the growth rate of *Nitzschia lecointei* (Torstensson et al., 2013).

But limited studies have investigated interactions between OA and combination of temperature and photoperiod (i.e.

conditions in different seasons) on diatoms. *Skeletonema costatum* is a widespread, eurythermal and euryhaline diatom species, which frequently causes red tide. We hypothesized the effect of OA on *S. costatum* may be modulated by photoperiod and temperature. In the present study, we investigate the physiological performance of marine diatom

*Skeletonema costatum* under two different $CO_2$ levels and three combinations of temperature and photoperiod, which simulated different seasons in typical temperate oceans (winter, 5 ℃ with 8:16 L:D; spring or autumn, 15 ℃ with 12:12 L:D; summer, 25 ℃ with 16:8 L:D).

## 2 Materials and methods

### 2.1 Culture conditions

The diatom *Skeletonema costatum* in this study was isolated from Gaogong Island, Lianyungang, Jiangsu province (34°70′74.95″N, 119°49′26.47″E). Before being used in experiments, the cells were cultured in autoclaved natural seawater enriched with f / 2 medium (Guillard and Ryther, 1962). Semi-continuous cultures were maintained in 500 ml Erlenmeyer flasks with a filter unit (Millex GP, Merck, USA) in order to aerate sterile air. Triplicate independent cultures were set for each treatment at the light intensity of 150 μmol photons $m^{-2}$ $s^{-1}$.

### 2.2 Experimental setup

In order to evaluate effects of $pCO_2$ levels and different combination of temperature and photoperiod on *S. costatum*, cells were cultured under winter (5 ℃ with light: dark cycle of 8:16 h), spring / autumn (15 ℃ with 12:12 h), summer (25 ℃ with 16:8 h) conditions independently with two $pCO_2$ levels (400 ppm, LC; 1000 ppm, HC), simulating temperature and daylength conditions of different seasons in typical temperate oceans. Temperatures and light intensity (150 μmol photons $m^{-2}$ $s^{-1}$) were controlled by illumination incubators (GXZ-500B, Ningbo, China). Cells were inoculated in cultures with fresh medium which was aerated with ambient air (400 ppm) or $CO_2$-enriched air (1000 ppm). The high $pCO_2$ level was manipulated by a $CO_2$ plant incubator (HP 1000 G-D, Ruihua Instruments, Wuhan, China). Cultures were kept at exponential phase by diluting every 3 d, and cells concentrations were controlled below $2×10^5$ cell $ml^{-1}$ in order to minimize the effect of cell metabolism on carbonate chemistry in medium. The changes in culture pH was less than 0.05 during the 3 d (8.10 ± 0.01 for LC and 7.85 ± 0.01 for HC in winter; 8.14 ± 0.01 for LC and 7.85 ± 0.01 for HC in spring / autumn; 8.19 ± 0.02 for LC and 7.89 ± 0.02 for HC in summer). After acclimating to different treatments for at least 40 generations, following parameters were measured.

### 2.3 Growth measurement

To estimate the growth of *S. costatum*, triplicate samples (1 ml each) were collected from each treatment at 48 and 72 h after dilution and fixed with 10 μl Lugol's solution, then a plankton counting chamber (DSJ-01, Xundeng Instruments, Xiamen, China) was used to count cells directly under an optical microscope (DM500, Leica, Germany). The specific growth rate was

calculated as: $\mu = (\ln N_t - \ln N_0) / (t - t_0)$, where $N_t$ represents the cell concentration (cells $ml^{-1}$) at time t; $N_0$ represents the cell

concentration at time $t_0$, $t - t_0 = 1$ d. The growth rates were averaged from 3 dilution processes within each growth condition.

## 2.4 Chlorophyll *a* and BSi measurements

Samples were filtered onto GF / F filters (25 mm, Whatman, UK), and chlorophyll *a* were extracted with 4 ml of methanol at

4 °C for 24 h in darkness. An ultraviolet spectrophotometer (Ultrospect 3300 pro, Amersham Bioscience, Sweden) was used

to detect the absorption values of supernatant under 632 nm, 665 nm and 750 nm after centrifuging (Biofuge primo R,

Thermo, Germany). The chlorophyll *a* concentration (pg $cell^{-1}$) of *S. costatum* was calculated by the equation of Ritchie

(2006).

Samples (200 ml) for biogenic silica (BSi) measurement (pmol $cell^{-1}$) were filtered onto polycarbonate filters (0.8 μm, Merck

Millipore, Germany) by polysulfone filter funnel (25 mm, Pall Corporation, UK), and filters were then dried at 80°C for 24 h.

BSi on the filter was digested by 4 ml of 0.2 M NaOH in boiling bath for 40 min, and were neutralized with 1 ml of 1M HCl

when cooled. The supernatant (1 ml) was diluted with 4 ml of milli-Q water, and then 2 ml of molybdate soln and 3 ml of

reducing agent were added into tubes. The absorption was measured at 810 nm by an ultraviolet spectrophotometer

(Ultrospect 3300 pro, Amersham Bioscience, Sweden) after the color developing for 2-3 h (Brzezinski and Nelson, 1995).

## 2.5 Photosynthesis and respiration measurements

The net photosynthetic rate under culture condition, photosynthetic oxygen evolution rate versus light intensity (P-I) curve

and photosynthesis versus DIC concentration (P-C) curve of *S. costatum* were measured through a Clark-type oxygen

electrode (Oxygraph+, Hansatech, UK), in which temperature was controlled by a thermostatic water bath (DHX-2005,

China).

For measurement of net photosynthesis under culture condition, light intensity was set as 150 μmol photons $m^{-2}$ $s^{-1}$(light

intensity during culture), provided by a halogen lamp (QVF135, Philips, Netherlands). Sample of 50 ml was filtered (< 0.02

MPa) onto a cellulose acetate membrane (Xinya Instruments, Shanghai, China). Due to the shorter measurement time than

that of P-C curve, cells were filtered and resuspended in 5 ml pre-aerated fresh medium (without buffer) under cultured

condition, which was then used to determine oxygen evolution rate and cell concentration. Oxygen consumption was

measured under darkness which was realized by covering an opaque box on the reaction chamber.

**2.6 P-C curve measurement**

For P-C curve, photosynthetic oxygen evolution was measured under same light intensity as culture. Sample of 50 ml was

filtered and resuspended in 5 ml fresh $C_i$-free f / 2 tris buffered medium (pH 8.12), which was then used to determine oxygen

evolution rate and cell concentration. Same pH level (pH 8.12) was set for both LC and HC treatments in order to compare $K_m$ and $V_{max}$ at same percentages of DIC species. The remaining intracellular $C_i$ was depleted by exposing cells to culture light intensity for 20-30 min. Net photosynthetic rate was determined after adding calculated amounts of $NaHCO_3$ stock solution (0.025 M and 0.2 M). The final concentrations of DIC were 0, 0.025, 0.05, 0.1, 0.2, 0.4, 1, 2 and 4 mM. Net photosynthetic rates under different DIC concentration were fitted by Michaelis-Menten equation.

## 2.7 P-I curve measurement

For P-I curve, oxygen consumption rates in darkness and net photosynthetic oxygen evolution at 7 different light intensities (0, 10, 20, 50, 100, 200, 500, 1000 µmol photons $m^{-2}$ $s^{-1}$) were identified. Light intensity was achieved by adjusting the distance between the halogen lamp and oxygen electrode chamber. Cells were also filtered and resuspended in pre-aerated fresh medium in a similar way to photosynthesis measurement under culture condition. Photosynthetic rate and light intensity data were fitted according to Henley (1993): $P = P_m \times \tanh{(\alpha \times PAR / P_m)} + R_d$, where PAR is irradiance, P is photosynthetic rate, $P_m$ is light-saturated photosynthetic rate, $\alpha$ is initial slope of P-I curve, $R_d$ is dark respiration rate. $I_k$ (saturating irradiance for photosynthesis) and $I_c$ (light compensation point) were also calculated by: $I_k = P_m / \alpha$, $I_c = R_d / \alpha$.

Cells were measured under 8 different PAR levels (0, 10, 20, 50, 100, 200, 500, 1000 µmol photons $m^{-2}$ $s^{-1}$) lasting for 10 s each using a hand-held fluorometer (AquaPen-C AP-P 100, Chech) for rapid light curves (RLCs) measurement. Relative electron transport rates (rETR) of *S. costatum* were measured, and were estimated as Wu, et al. (2010): $rETR = PAR \times Y (II) \times 0.5$, where PAR represents the photon flux density of actinic light; Y(II) represents the effective quantum yield of PSII, 0.5 is based on the assumption that PSII receives half of all absorbed quanta. RLCs were fitted as: $P = PAR / (a \times PAR^2 + b \times PAR + c)$, where P represents rETR; a, b and c are model parameters. The relative photoinhibition ratio of rETR was calculated as: $Inh (\%) = (rETR_{max} - rETR_x) / rETR_{max} \times 100 \%)$, where $rETR_x$ is the value of rETR at 1000 µmol photons $m^{-2}$ $s^{-1}$.

## 2.8 Protein measurements

Ribulose-1,5-bisphosphate carboxylase/oxygenase (RubisCO) large subunit binding protein (RbcL) and PSII proteins (PsbA (D1), PsbD (D2), and PsbB (CP47)) were measured under different $pCO_2$ levels and combinations of temperature and daylength. D1 and D2 proteins are located in reaction center, while CP47 is a junction of antenna, and RbcL is a component of RubisCO which is a key enzyme in the $CO_2$ fixation process. For the relative value of protein measurements, cells were filtered and resuspended in 2 ml of extracting medium (50 mM Tris-HCl, pH 7.6, 5.0 mM $MgCl_2$, 10 mM NaCl, 0.4 M sucrose, and 0.1 % BSA) according to Ma et al. (2019). After cell disruption and centrifugation, supernatant liquid was used to measured chlorophyll concentration (µg $ml^{-1}$) according to Arnon (1949): $C = D_{652} \times 1000 \div 34.5 \times T$, where C represents total chlorophyll concentration, $D_{652}$ represents absorption value in 652 nm, T represents dilution ratio, 1000 and 34.5 are

constants. Same concentration of chlorophyll (2.4 micrograms) per lane was used for 12 % sodium dodecyl sulfate-polyacrylamide gel electrophoresis (SDS-PAGE, Mini PROTEAN, Bio-rad, America) at 150 V for 1 h, and the proteins were transferred into polyvinylidene difluoride (PVDF) membranes which were then immersed in blocking solution with antibodies (D1, D2, CP47, RbcL and Actin; Agrisera) for 1 h, and successively, goat anti-rabbit secondary antibodies were used. Blots were developed by using enhance chemluminesence luminescence (ECL) reagent and were quantified with a chemiluminescence detection system (Tanon 5500, Shanghai, China). Actin was used as internal control in order to correct the experimental error in the process of quantitative sample loading of protein, to ensure the accuracy of the experimental results. And the data provide us with a general trend, not accurate concentrations, among different treatments.

**2.9 Data analyses**

Data were analyzed with IBM SPSS Statistics 24 and are presented as mean ± SD (standard deviation). One-way ANOVA was used to compare differences among combination of temperature and photoperiod treatments. The independent-samples *t*-test was applied to compare differences between two $pCO_2$ levels. General linear model was conducted to assess the interactive effects of $CO_2$ level and combination of temperature and photoperiod on growth rate, rETR, photosynthesis and respiration, contents of chlorophyll *a* and BSi and proteins. When *P* values were under 0.05, tukey test was used for *post hoc* analysis.

**3 Results**

**3.1 Specific growth rate**

Growth rate of *S. costatum* raged from $0.47 \pm 0.01$ to $3.22 \pm 0.08$ $d^{-1}$ under different treatments and increased significantly with increasing temperature and daylength ($P < 0.05$) regardless the $pCO_2$ level (Fig. 1). In summer season, elevated $pCO_2$ showed no significant effects, however, it remarkably influenced the growth rate in other seasons. Elevated $pCO_2$ enhanced the growth rate by 11 % in spring and autumn ($P < 0.001$), while the reverse pattern was found in winter ($P < 0.001$). General linear model indicated that the season and $CO_2$ level had a notable interaction on specific growth rate ($P < 0.001$, Table 1).

**3.2 Chlorophyll *a* and BSi contents**

Under ambient $CO_2$ condition, chlorophyll *a* content was enhanced by increased temperature and daylength (Table 2), and the content was 22 % higher in summer compared with winter ($P = 0.008$). When $CO_2$ was elevated, chlorophyll *a* content in winter was 42 % and 32 % lower than that in spring and summer respectively ($P = 0.001$, $P = 0.004$). Elevated $pCO_2$

decreased chlorophyll *a* content in winter ($P = 0.022$) while enhanced it in spring ($P = 0.002$) and had no significant impact in summer. A significant interaction between season and $CO_2$ can be found ($P < 0.001$) (Table 1).

A different trend was detected for BSi content (Table 2). Under ambient $pCO_2$, BSi content decreased with higher temperature along with longer daylength and the value in winter was significantly higher than that in spring and summer ($P=0.005, 0.002$ respectively). Higher $pCO_2$ decreased BSi significantly in winter ($P = 0.016$) and spring ($P = 0.007$) while had no significant influence on the content in summer ($P = 0.3$). There is a significant interaction between season and $CO_2$ on BSi content ($P < 0.05$) (Table 1).

### 3.3 Photosynthesis and respiration

Net photosynthetic oxygen evolution and dark respiration rates showed similar patterns under same $CO_2$ condition (Fig. 2). The lowest photosynthesis and respiration rates were observed under winter condition, and maximal rates were observed in summer at each $pCO_2$ level. Both photosynthesis and respiration rates increased with increasing temperature and daylength ($P < 0.05$). Elevated $pCO_2$ inhibited net photosynthetic rate under winter condition ($P = 0.0013$) while photosynthesis was enhanced by elevated $pCO_2$ in spring and autumn ($P = 0.006$). In addition, higher $pCO_2$ stimulated dark respiration rate in spring and autumn ($P < 0.001$). Both photosynthesis and respiration were not impacted by higher $pCO_2$ in summer. Interaction between season and $CO_2$ on net photosynthetic rate was detected ($P = 0.035$). Positive relationships of dark respiration or net photosynthetic rates and growth rate were observed (Fig. 5a, b).

### 3.4 P-C curve

Combination of temperature and photoperiod in different seasons had a significant effect on $V_{max}$ ($P < 0.001$, Table 1, Fig. 3b), cells under spring / autumn season condition had the highest $V_{max}$ at both $pCO_2$ levels, but the lowest $V_{max}$ was found in winter at elevated $pCO_2$ and in summer at ambient $pCO_2$. Elevated $pCO_2$ significantly enhanced the values in higher temperature and longer daylength treatments (129 % for spring / autumn, 130 % for summer when compared with ambient $CO_2$ level). With respect to $K_m$, different season had no significant effect on it at ambient $pCO_2$, while $K_m$ in summer was lowest among the seasons at elevated $pCO_2$ (70 % and 69 % lower than winter and spring / autumn respectively). Elevated $pCO_2$ had significant influences on $K_m$ under same season besides summer and the values increased by 143 % and 113 % in winter and spring / autumn respectively when compared with low $pCO_2$ ($P = 0.021$, $P = 0.003$ in winter and spring / autumn). Interaction between $CO_2$ and combination of temperature and photoperiod on parameters of P-C curves were detected (Table 1).

### 3.5 P-I curve

Net photosynthetic oxygen evolution rates increased with increasing light intensity initially and then reached the plateaus in all seasons and the curves under winter condition reached the plateaus much earlier than the other two seasons (Fig. 4). Higher temperature and prolonged daylength had a main effect on $P_{max}$, $R_d$, $I_k$ and $I_c$. However, elevated $pCO_2$ instead of season had main effect on $\alpha$ (Table 1). $P_{max}$ was enhanced when temperature increasing with prolonged daylength ($P < 0.05$) except when the summer season was compared with spring and autumn condition at elevated $pCO_2$ (Table 3). Effects of elevated $pCO_2$ was only observed in spring and autumn ($P = 0.03$). $I_k$ increased remarkably in summer under both $pCO_2$ treatments which was similar as $R_d$ at elevated $pCO_2$ ($P < 0.05$). There was a significant interaction between $CO_2$ and combination of temperature and photoperiod on $R_d$ (Table 1).

At higher $pCO_2$ level, $rETR_{max}$ values were significantly different among different seasons ($P < 0.05$), and the highest value was found in spring and autumn (52 % and 14% higher than winter and summer, Table 3). Elevated $pCO_2$ decreased $rETR_{max}$ in winter and summer ($P < 0.001$ in winter, $P = 0.01$ in summer). $rETR_{max}$ had positive correlation with growth rate when temperature and daylength increased from winter to spring / autumn condition. However, as temperature and daylength continuously increasing from spring / autumn to summer, the correlation became negative (Fig. 5c). Interactions between the two factors were detected on $rETR_{max}$ ($P < 0.001$, Table 1). Photoinhibitions were found in RLCs of cells under all treatments. As the season processed from winter to summer, photoinhibitions were alleviated significantly in both $pCO_2$ levels ($P < 0.05$). In spring and autumn, the inhibition at higher $pCO_2$ was significantly decreased compared with ambient $pCO_2$ condition ($P = 0.039$, Table 3).

### 3.5 PSII protein concentrations

The relative value of RbcL and key PSII proteins (PsbA (D1), PsbD (D2), and PsbB (CP47)) were quantified in different seasons under ambient and elevated $CO_2$ conditions (Fig. 6). At ambient $pCO_2$ level, the highest contents of all four proteins were detected in spring and autumn. Elevated $pCO_2$ significantly enhanced RbcL, D1 and D2 protein contents in winter ($P < 0.05$). In addition, higher $pCO_2$ led to increase of CP47 in summer ($P = 0.006$).

### 4. Discussion

Phytoplankton like diatoms have already evolved several strategies to cope with different temperature and daylength in temperate oceans, where variations of season are evident. However, the ongoing elevated $pCO_2$ combining with changes in temperature and daylength is a new stress on diatoms and we know little about their interactions. Therefore, we examined the combined effects of $pCO_2$ and seasonal changes in temperature and photoperiod on the physiological performance of a typical marine diatom *S. costatum*.

### 4.1 Physiological responses of *S. costatum* to different combinations of temperature and photoperiod

In the present study, the growth rate of *S. costatum* increased with increasing temperature and daylength regardless of the $pCO_2$ level (Fig. 1). Previous studies showed that most phytoplankton, such as *Chlamydomonas reinhardtii*, *Trichodesmium* or *Alexandrium catenella*, grew faster under prolonged photoperiods (Cai and Gao, 2015) although *Alexnadrium minutum* grew faster under shorter photoperiods (Wang et al., 2019). For *S. costatum*, a remarkably higher contribution of $HCO_3^-$ to the overall carbon uptake was observed under light dark cycles compared with continuous light, and a shorter photoperiod led to lower photosynthetic affinity for inorganic carbon (Rost et al., 2006). Basically, the Chl *a* quota in microalgae increases with decreasing daylength, however, our results exhibited inverse pattern (Table 2). The inconsistency might be caused by the different temperatures set in studies, which is another main environmental factor affecting the growth of diatoms.

Increasing temperature may lead to various changes in growth rate depending on whether the temperature is optimal for the species. For *S. costatum*, its growth rate has been shown to increase with temperature up to 30 ℃ (Ebrahimi and Salarzadeh, 2016). Zhang et al. (2020) also found that the growth rate of *S. costatum* could increase with temperature from 5 ℃ to 30 ℃ and then drop sharply. The underlying mechanism is that elevated temperature promotes *S. costatum* metabolic rates when nutrients are abundant. This could be shown by the relationship between respiration and growth. Higher mitochondrial respiration can result in higher growth rate theoretically (Geider and Osborne, 1989), since this process provides ATP and carbon skeletons (Raven et al., 2017). It seems that temperature shows the dominant effect compared with daylength.

Rubisco is an important enzyme for carbon fixation, psychrophilic diatoms utilize increasing abundance of RbcL protein to maintain high cellular enzymatic rates and growth rate at low temperature (Young et al., 2015). However, for diatoms in temperate area, the amount of RbcL protein decreased in low temperature along with short daylength compared with higher temperature and longer daylength in our results. Low abundance of RbcL was in line with slower growth rate in winter condition.

## 4.2 Effect of ocean acidification under different seasons

Our results showed that the impacts of elevated $pCO_2$ on *S. costatum* depended on seasonal changes in temperature and photoperiod. High $pCO_2$ had enhanced the growth of *S. costatum* in spring / autumn and reduced it in winter, while no significant effects were detected in summer (Fig. 1). $CO_2$ concentration mechanisms (CCMs) is energy-dependent and high $pCO_2$ down-regulate CCMs of most phytoplankton including *S. costatum*, so the saved energy could be used for growth (Raven et al., 2017). Higher initial slope of the P-I curve at elevated $pCO_2$ might be partly responsible for the higher growth rate compared with that at ambient $pCO_2$ (Table 3). But in winter, growth decreased under OA condition, although photosynthesis and respiration had no significant changes. This is because the combination of biochemical and biophysical CCMs may cause the lack of a positive response to elevated $pCO_2$ under near-optimal growth conditions (Passow, 2015). In addition, when other environmental factors are stressful, the sensitivity of diatoms to $CO_2$ and temperature is prominent (Taucher et al., 2015). The shorter daylength and low temperature simulating winter condition in present study can be seen as

stressors, under which *S. costatum* was more sensitive to elevated $pCO_2$. When temperature and photoperiod are optimal, positive or neutral effects of higher $pCO_2$ were observed. However, different patterns were reported for other species, such as *E. huxleyi* and the macroalgae *Ulva linza* (Bretherton et al., 2019; Yue et al., 2019). For these two species, reduced growth rate at elevated $pCO_2$ were found when the daylength was longest. High temperature might accelerate nutrient uptake and metabolic rates, which may alleviate the negative effects of longer daylength under higher $pCO_2$ environment (Bretherton et al., 2019; Yue et al., 2019). Maximum photosynthetic rate increased significantly under higher $pCO_2$ in spring and autumn condition. This was consistent with the higher photosynthetic efficiency and growth rate (Table 3, Fig. 1). Meanwhile, $K_m$ increased under high $CO_2$ concentration which means the affinity for $CO_2$ decrease and thus CCMs are down-regulated (Fig. 3).

Silicification directly relates to cell division and growth, and is independent of photoperiod (Brzezinski, 1992). BSi contents generally increased with decreasing growth rate when any limiting factors such as temperature, light or ammonium exist (Martin et al., 2000). In the present study, elevated $pCO_2$ mitigated the negative effects of temperature and photoperiod limitation on BSi content (Table 2). Higher BSi contents in winter under ambient $CO_2$ condition can intensify the ballasting effects and thus impact the sinking rate of organic matters produced by diatoms.

Lavaud et al. (2016) indicated that PSII activity and phosphorylation of thylakoid protein may play a crucial role in controlling the change of the photosynthetic activity. Higher $pCO_2$ induced higher photosynthetic efficiency in PSII, with increased contents of D1 and D2 proteins in winter (Fig. 6). Proteins will be degraded and synthesized rapidly after damage. Therefore, the decline in growth under winter condition might result from the increased metabolic costs of photoprotection and elevated D1 turnover under the combination of short daylength limitation and low temperature (Hoppe et al., 2015). Although RbcL decreased with elevated $CO_2$ in some studies that might be caused by the decrease in RuBP concentration for diatoms (Endo et al., 2015), McCarthy et al. (2012) observed an increase in Rubisco concentration with higher $CO_2$, which is in line with the present study under winter condition. Light and temperature could affect RbcL amount in phytoplankton. RbcL increased slightly with higher $CO_2$ at low light intensity condition, while it decreased slightly at higher light intensity in the research of Levitan et al. (2010). In the present study, the combination of low temperature and short daylength might lead to a complex trend of RbcL, elevated $CO_2$ might stimulate the sensitivity of *S. costatum* to low temperature which lead to a steep rise of RbcL in winter.

For diatoms, a mounting body of studies has pay attention to the effect of interactions of ocean acidification and other environmental factors such as light intensity, UV, temperature, nutrient limitation, salinity or photoperiod (Gao et al., 2012; Yue et al., 2019), and the results showed positive, negative or neutral effects (Xu et al., 2014c; Li et al., 2017). However, few studies combined elevated $pCO_2$ with seasonal changes in seawater physical and chemical characters on marine diatoms. In this study, temperature and photoperiod were chosen as seasonal factors to investigate the combined effects of $pCO_2$ and these two factors on physiology of *S. costatum*. Our results suggested temperature and photoperiod could mediate effects of elevated $pCO_2$ on the typical diatom *S. costatum*. Positive effects of OA on growth and photosynthesis were observed in spring and autumn, while negative effects were found in winter condition. To better understand how global climate changes

would affect marine diatoms in the future, it is necessary to explore the interactive effects of ocean acidification with seasonal changes in seawater characters.

## Author Contributions

JX and FL conceived and designed the experiments; HL and TX carried out the experiments; HL, FL and JX analyzed data; HL wrote the draft of the paper; FL, JM and JX revised the manuscript and approved this version for submission.

## Competing interests

The authors declare that they have no conflict of interest.

## Acknowledgements

This study was funded by China Postdoctoral Science Foundation (2019M661766), the Six Talents Peaks in Jiangsu Province (JY-086), Postgraduate Research & Practice Innovation Program of Jiangsu Province (KYCX19_2290), the Priority Academic Program Development of Jiangsu Higher Education Institutions.

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

Table 1: Significance test results of growth, BSi, photosynthetic and respiration rates, parameters of P-I ($P_{max}$ is the maximum net photosynthetic rate, α is the photosynthetic efficiency, $R_d$ is the dark respiration rate), P-C ($K_m$ is half-saturation constant and $V_{max}$ is the inorganic carbon-saturated maximal rate of photosynthesis) and RLCs curves ($rETR_{max}$, which is the maximum relative electron rate) for combination of temperature and photoperiod (season), $CO_2$, and their interactions.

| Parameter | Season | | $CO_2$ | | Season * $CO_2$ | |
|---|---|---|---|---|---|---|
| | F | p | F | p | F | p |
| Specific growth rate | 7662.1 | **<0.001** | 0.3 | 0.569 | 27.3 | **<0.001** |
| BSi | 22.6 | **<0.001** | 22.5 | **<0.001** | 10.9 | **<0.001** |
| Photosynthesis | 452.1 | **<0.001** | 0.1 | 0.735 | 4.5 | **0.0345** |
| Dark respiration | 51.9 | **<0.001** | 7.6 | **0.018** | 1.9 | 0.182 |
| $P_{max}$ | 85.5 | **<0.001** | 6.3 | **0.028** | 3.4 | 0.069 |
| α | 1.8 | 0.211 | 6.2 | **0.028** | 1.5 | 0.262 |
| $R_d$ | 7.3 | **0.010** | 3.3 | 0.097 | 4.2 | **0.044** |
| $V_{max}$ | 88.4 | **<0.001** | 136.8 | **<0.001** | 49.9 | **<0.001** |
| $K_m$ | 20.5 | **<0.001** | 35.7 | **<0.001** | 6.7 | **0.011** |
| $rETR_{max}$ | 85.3 | **<0.001** | 98.5 | **<0.001** | 26.0 | **<0.001** |

Table 2: Chl $a$ (pg cell$^{-1}$) and BSi (pmol cell$^{-1}$) contents of *S. costatum* acclimated to ambient and elevated $pCO_2$ in different seasons. The data are mean ± SD values of triplicate cultures (n = 3). Different lowercases represent significant differences ($P < 0.05$) between two $CO_2$ levels under same season ($t$-test).

| Treatments | Chl $a$ | | BSi | |
|---|---|---|---|---|
| | LC | HC | LC | HC |
| Winter | 0.18 ± 0.008[a] | 0.15 ± 0.013[b] | 0.035 ± 0.003[a] | 0.025 ± 0.003[b] |
| Spring / Autumn | 0.19 ± 0.007[a] | 0.26 ± 0.014[b] | 0.025 ± 0.002[a] | 0.019 ± 0.001[b] |
| Summer | 0.22 ± 0.015[a] | 0.23 ± 0.024[a] | 0.023 ± 0.001[a] | 0.024 ± 0.002[a] |

Table 3: Photosynthetic parameters fitted from $P_{(O2)}$-I and rapid light curves for *S. costatum* acclimated to ambient and elevated $pCO_2$ in different seasons. $P_{max}$ (pmol $O_2$ cell$^{-1}$ h$^{-1}$) is the maximum net photosynthetic rate, α is the photosynthetic efficiency, $R_d$ (pmol $O_2$ cell$^{-1}$ h$^{-1}$) is the dark respiration rate, $I_k$ (μmol photons m$^{-2}$ s$^{-1}$) is the photosynthetic saturated light intensity, $I_c$ (μmol photons m$^{-2}$ s$^{-1}$) is light compensation point, $rETR_{max}$ is the maximum relative electron rate and Inh is the relative photoinhibition ratio of rETR. Different lowercases represent significant differences ($P < 0.05$) between two $CO_2$ levels under same season ($t$-test).

| | $P_{max}$ | α($O_2$) | $R_d$ | $I_k(O_2)$ | $I_c$ | $I_k$ (ETR) | α (ETR) | $rETR_{max}$ | Inh (%) |
|---|---|---|---|---|---|---|---|---|---|
| Winter-LC | 0.13 ± 0.01[a] | 0.0012 ± 0.00007[a] | 0.011 ± 0.006[a] | 112.4 ± 17.3[a] | 9.8 ± 5.1[a] | 302.7 ± 8.3[a] | 0.21 ± 0.008[a] | 59.0 ± 1.76[a] | 147.3 ± 6.7[a] |
| Winter-HC | 0.11 ± 0.02[a] | 0.0012 ± 0.00045[a] | 0.003 ± 0.002[a] | 97.5 ± 21.7[a] | 7.2 ± 3.3[a] | 193.6 ± 10.7[b] | 0.22 ± 0.005[a] | 40.3 ± 1.43[b] | 135.0 ± 16.8[a] |
| Spring-LC | 0.23 ± 0.04[a] | 0.0011 ± 0.00012[a] | 0.007 ± 0.003[a] | 204.1 ± 28.0[a] | 5.7 ± 1.9[a] | 373.0 ± 8.5[a] | 0.20 ± 0.0005[a] | 67.4 ± 1.59[a] | 68.7 ± 6.9[a] |
| Spring-HC | 0.31 ± 0.03[b] | 0.0016 ± 0.00019[b] | 0.019 ± 0.007[a] | 192.3 ± 19.9[a] | 11.1 ± 3.5[a] | 386.6 ± 10.2[a] | 0.21 ± 0.005[b] | 61.6 ± 3.77[a] | 54.5 ± 4.3[b] |
| Summer-LC | 0.35 ± 0.05[a] | 0.0011 ± 0.00009[a] | 0.016 ± 0.008[a] | 336.7 ± 66.3[a] | 14.5 ± 7.1[a] | 465.1 ± 18.6[a] | 0.12 ± 0.018[a] | 57.1 ± 0.04[a] | 28.9 ± 4.3[a] |
| Summer-HC | 0.41 ± 0.05[a] | 0.0013 ± 0.00018[a] | 0.032 ± 0.011[a] | 322.8 ± 36.1[a] | 25.4 ± 10.5[a] | 462.3 ± 16.8[a] | 0.16 ± 0.009[b] | 53.8 ± 1.29[b] | 27.1 ± 2.7[a] |

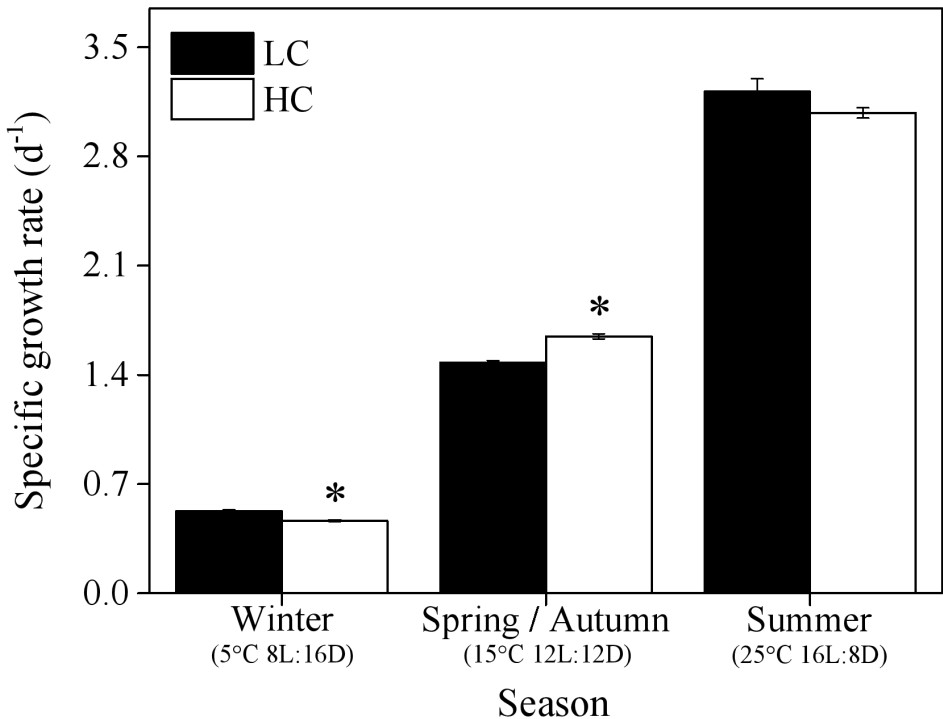

**Figure 1: Specific growth rate of *S. costatum* acclimated to ambient (LC, black bars) and elevated pCO₂ (HC, white bars) under different combination of temperature and photoperiod conditions. The data are mean ± SD values of triplicate cultures (n = 3). Asterisks represent significant differences (*P* < 0.05) between two CO₂ levels under same season condition (*t*-test).**


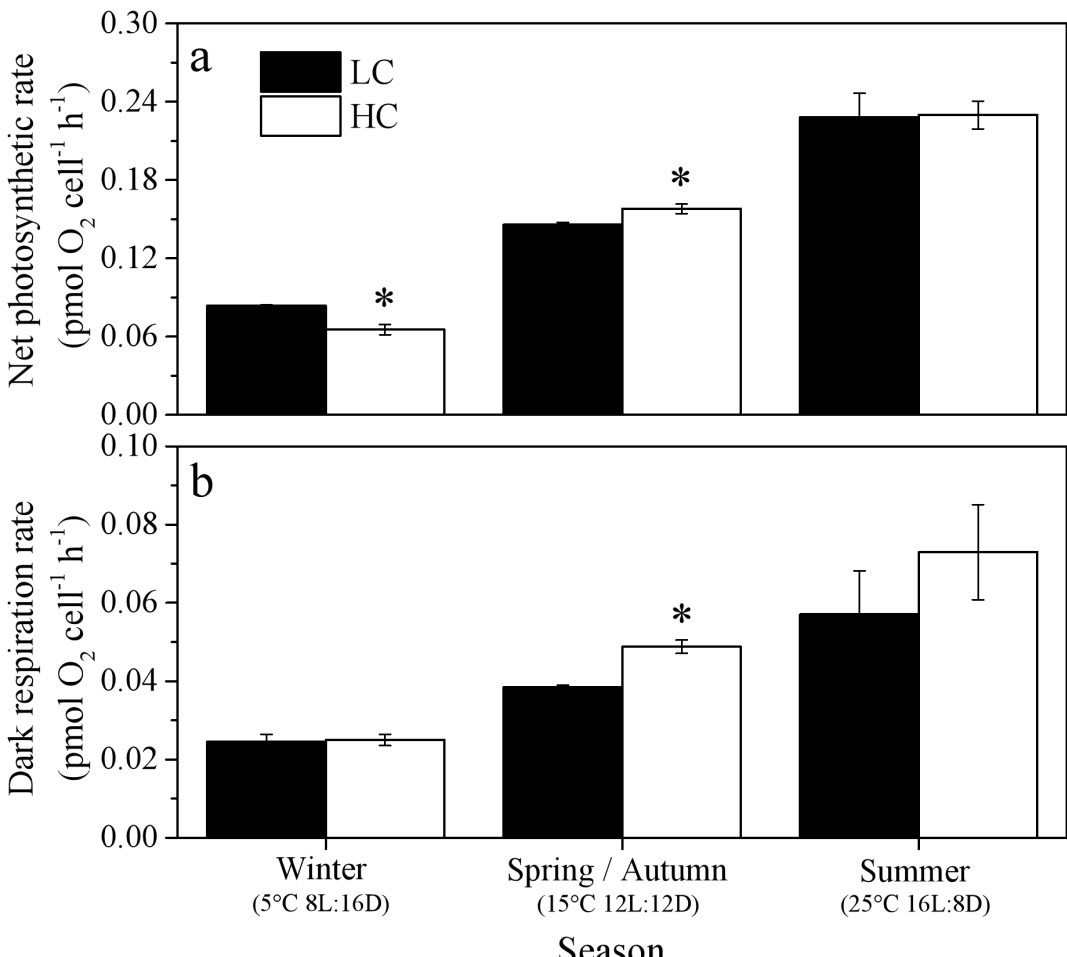

**Figure 2: Net photosynthetic (a) and dark respiration rates (b) of *S. costatum* acclimated to ambient (LC, black bars) and elevated pCO₂ (HC, white bars) under different combination of temperature and photoperiod conditions. The data are mean ± SD values of triplicate cultures (n = 3). Asterisks represent significant differences ($P < 0.05$) between two CO₂ levels under same season condition (*t*-test).**


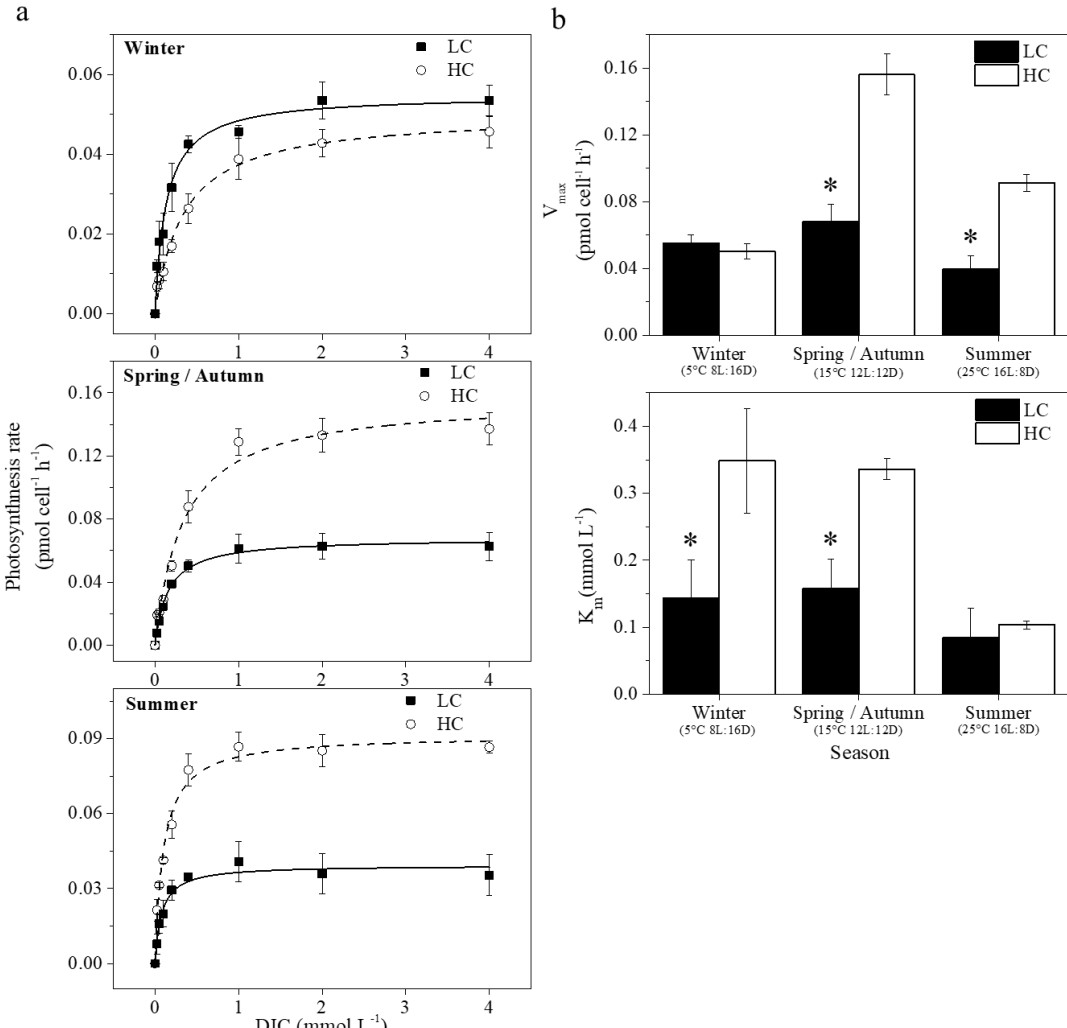

**Figure 3: (a) Photosynthesis-inorganic carbon (P-C) curves of cells acclimated to ambient and elevated $pCO_2$ in different seasons. (b) Photosynthetic parameters of P - C curves at ambient (LC, black bars) and elevated $pCO_2$ (HC, white bars) in different seasons. $K_m$ is half-saturation constant and $V_{max}$ is the inorganic carbon-saturated maximal rate of photosynthesis. The data are mean ± SD values of triplicate cultures (n = 3). Asterisks represent significant differences ($P < 0.05$) between two $CO_2$ levels under same season (*t*-test).**


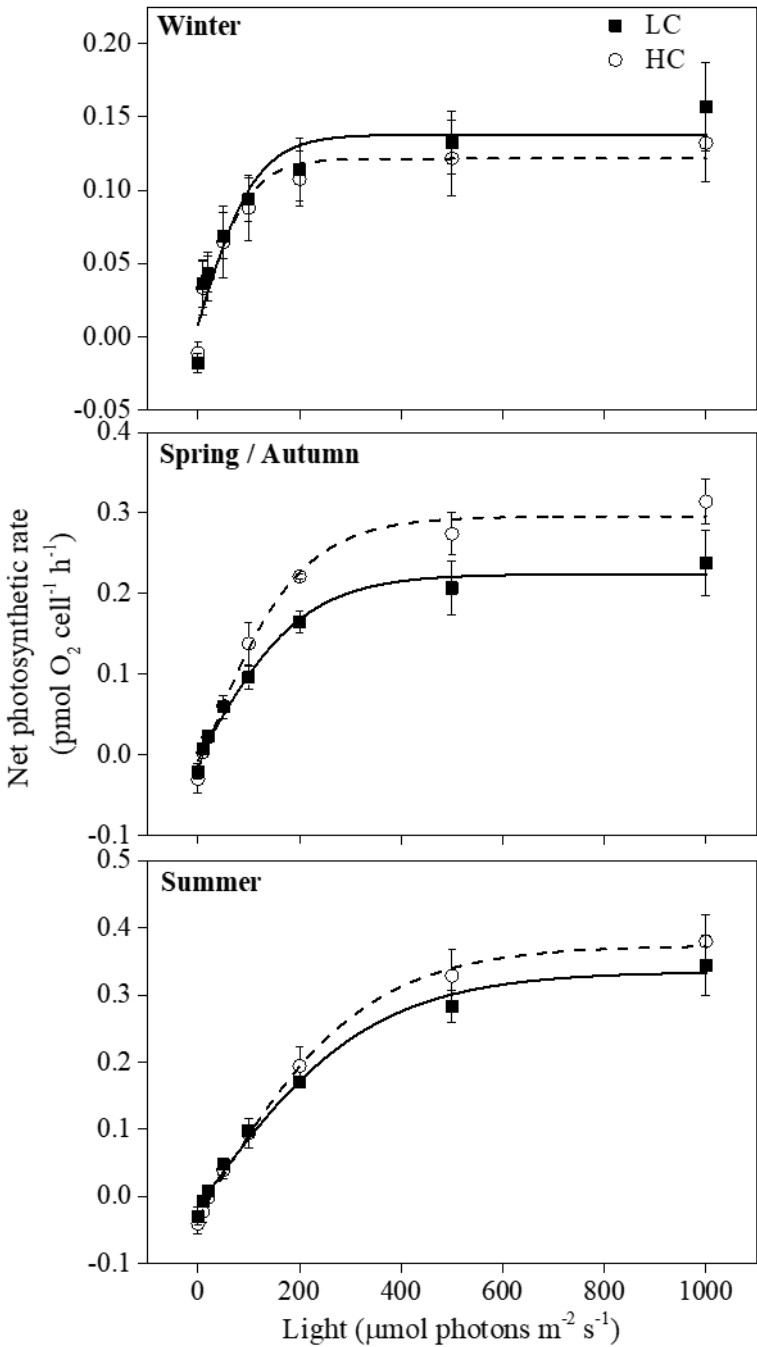

**Figure 4: Photosynthesis-light curves (P-I curves) of cells acclimated to ambient and elevated pCO₂ in different seasons. The data are mean ± SD values of triplicate cultures (n = 3).**

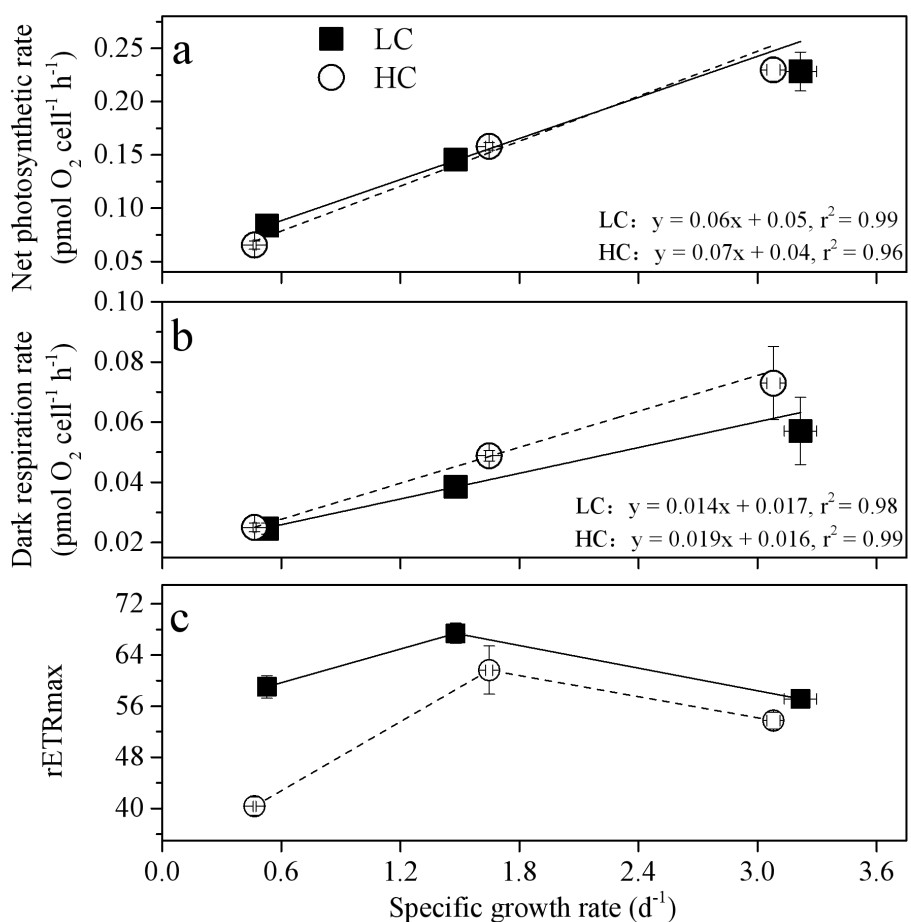


**Figure 5: The relationship between net photosynthetic rate (a), dark respiration (b), rETR$_{max}$ (c) and specific growth rate of *S. costatum* acclimated to ambient (LC, black square) and elevated pCO$_2$ (HC, white circle) under different combination of temperature and photoperiod conditions. The data are mean ± SD values of triplicate cultures (n = 3).**

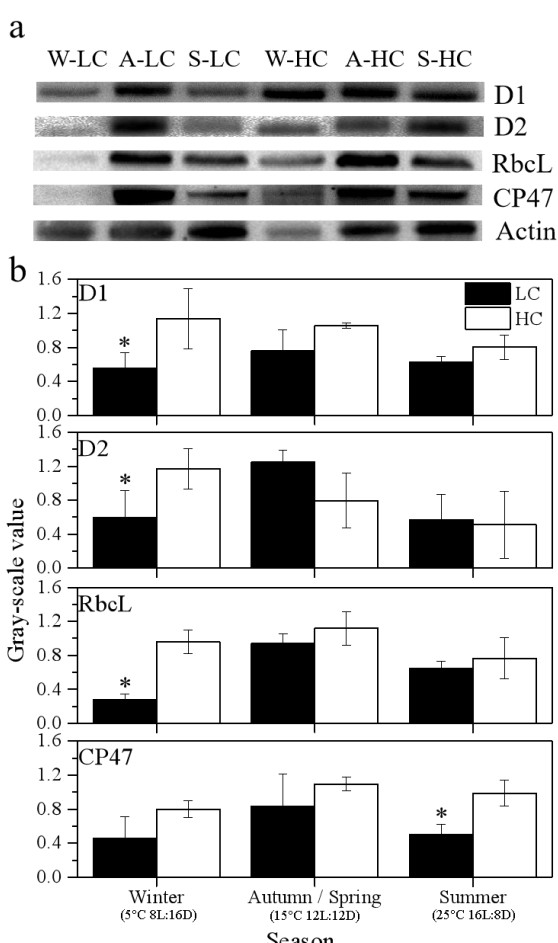

**Figure 6: (a) Immunoblot using antibodies against RbcL and key PSII proteins (PsbA (D1), PsbD (D2), and PsbB (CP47)) isolated from *S. costatum* acclimated to ambient and elevated pCO₂ in different season levels (W for winter, A for autumn, S for summer). Each line was loaded with similar amounts of proteins. (b) Quantitative analysis of proteins. The relative abundance of each band was estimated by densitometric scanning of the exposed films. Asterisks represent significant differences ($P < 0.05$) between two CO₂ levels under same season (*t*-test).**

