# Peer review of "Physiological responses of *Skeletonema costatum* to the interactions of seawater acidification and combination of photoperiod and temperature"

_Biogeosciences, 2020_

## Referee Comment (RC1) · Douglas Campbell (Referee) · 24 Aug 2020

bg-2020-303

I am very happy to see a study of the interaction of increased $pCO_2$ with temperature and photoperiod on a model diatom. These multifactorial experiments are challenging but necessary. To limit the (infinite) range of possible combinations the authors matched photoperiod to expected seasonal temperature.

I think the data, as presented, is valuable, but under-analyzed. I offer some suggestions

for some cross plots of physiological performance at growth condition with achieved growth rate. What is the growth return on O2 evolution across the conditions?

Abstract: Good

Materials and Methods:

2.3 The authors need to describe whether the growth rates taken every 2 days were averaged within each growth condition, or whether the maximum measured growth rate for a condition is taken. The presented growth rates are fast for a diatom.

'2.7 Measurements of PSII Proteins' orå '2.7 PSII Protein Measurements'å

(current header is technically correct but is archaic usage).

Figures/Results; Figure 1: Amazing growth rate under summer conditions; the fastest I have every seen I think for a diatom.

Figure 2: Lovely data, congratulations. Very surprising switch of the OA effect in winter. Suggested additional plot: V at growth pCO2? After all, most of the curve as plotted is above even the OA range of pCO2. So a hypothetical Vmax may not be as important as the achieved V at growth conditions.

Then, I would suggested plotting growth rate vs. V at growth condition (pCO2 and light)

Figure 4: Without standard curves, be very cautious in interpretation of the immunoblotting data. The example blot result shows near-saturation (non-linear response of signal to target abundance) for many of the bands. So the Y axis dynamic range of the greyscale plots may be considerably compressed relative to the actual change in protein target abundance. Once a band is black, it cannot get any blacker.

It is also very surprising that in winter OA increased RbcL signal.

---

## Referee Comment (RC2) · Anonymous Referee #2 · 1 Oct 2020

The manuscript titled "Physiological responses of Skeletonema costatum to the interactions of seawater acidification and combination of photoperiod and temperature" described a research attempting to explore the impact of high pCO2 (or ocean acidification, OA) under different seasons (combination of photoperiod and temperature) on the diatom S. costatum. The experiments are well conceived, and methods are clearly presented. The most interesting observation is that high pCO2 (OA) does not uniformly impact S. costatum under different seasons: with somewhat negative impact on winter conditions. The authors showed the interesting observation, but the experimental

design and data quality can be improved.

Here are some questions and suggestions for the authors.

On the experimental setup, the authors stated that the cell culture pH did not change over 0.05 units in the 3d of one generation (section 2.3), so one very basic question is: what are the pH values and ranges for the six different conditions (i.e. three seasons and two pCO2)? Since the manuscript is about how OA impacts S. costatum differently during different seasons, the acidification information, which can be presented easily as pH, is very critical to this whole article, however, this information is missing.

Following the cell culture pH question, the authors measured the photosynthesis (P) vs DIC curve at pH 8.12 (section 2.5) and very likely they did the same with P-I curve. It would be better if the authors measure the responses under lower pH for high pCO2 treated cells, according to the high pCO2 (OA) conditions. It should be expected that the pH is lower under OA conditions, and S. costatum acclimated to OA conditions may not photosynthesize better under the experimental condition with higher pH (8.12). As a result, the presented P-I and P-DIC curves for S. costaum cultured in HC (OA conditions) may not reflect their real physiological status in terms of photosynthesis under OA conditions. Also please note that Tris buffer is known to change pH significantly with different temperature, so it is important to measure or calculate the pH at certain temperature.

In the results session, the authors only mentioned and cited Fig. 1, while Figure 2, 3, 4 are listed at the end of the manuscript, none of those was referred in the text. In the tables presented, Table 2 and Table 3 do not show any units.

In section 3.5 where the authors present the "PSII protein concentrations", RbcL was included as key PSII proteins. Such claim should be red-flagged, likewise the statement of "RbcL is related to the function of QA". Inclusion of RbcL in PSII proteins is also found in the Abstract and Discussion. The authors should make sure what RbcL really does with creditable citations before writing assumptions or conclusions about RbcL.

The description of methodology "Values of Actin were divided by other densitometric scanning values of protein to calculate Gray-scale values" should be modified to indicate the supposing meaning of normalizing density to Actin. With the data presented in Figure 4, panel (a) and panel (b) do not seem to agree with each other. The western blot data does not look like a representative of the statistical data. For example, in W-HC (winter high pCO2) condition, the D1 density is much higher than Actin (Fig 4a), so such value is greater than 2 if analyzed using ImageJ, however, the data presented in statistics showed a value very tightly close to 0.9 (Fig 4b). It would be nice if all raw data (immunoblots) are presented to support the statistics in Fig 4b.

Other minor concerns: Section 2.7, "1000 and 34.5 are constants", what are the units? Or at least provide the unit of "where C represents total chlorophyll concentration". The use of letter "C" is ambiguous in the text. C was also used as carbon in line 219: "C fixation". For the measurement of specific growth rate, more details on how data were collected would be helpful. It would be better to have the raw data, cell concentration vs time (days), presented. Line 229, "initial slop" should be "initial slope".

---

## Author Comment (AC1) · 26 Oct 2020

**Reviewer 1**

Comments:

I am very happy to see a study of the interaction of increased $pCO_2$ with temperature and photoperiod on a model diatom. These multifactorial experiments are challenging but necessary. To limit the (infinite) range of possible combinations the authors matched photoperiod to expected seasonal temperature.

I think the data, as presented, is valuable, but under-analyzed. I offer some suggestions for some cross plots of physiological performance at growth condition with achieved growth rate. What is the growth return on $O_2$ evolution across the conditions?

**Response:** We thank the reviewer for the useful comments and suggestions that have greatly improved this manuscript.

The following figure is the relationship between net photosynthetic rates (pmol $O_2$

cell$^{-1}$ h$^{-1}$) and specific growth rates (d$^{-1}$) and we will add the figure and the relationships between other physiological parameters and growth in the revised MS.

[Figure]

Abstract: Good

Materials and Methods:

2.3 The authors need to describe whether the growth rates taken every 2 days were averaged within each growth condition, or whether the maximum measured growth rate for a condition is taken. The presented growth rates are fast for a diatom.

**Response:** We are sorry for the ambiguous description. More details were added in the manuscript. Cultures were diluted every 3 days and cell concentration was counted two and three days after the dilution, and growth rates were calculated based on the concentrations. The data showed in Fig. 1 were averaged growth rates calculated for three times at different dilution days.

Diatoms have diverse cell sizes and thus growth rates of different species vary a lot.

For *Skeletonema costatum*, the average growth rate is around 1.0-2.0 $d^{-1}$ under normal (20°C, 100 μmol photons $m^{-2}$ $s^{-1}$, 12L: 12D) conditions (Balzano et al., 2011;

Sakshaug and Andresen, 1986), which accords with the growth rates (1.6 $d^{-1}$) under autumn condition (15°C, 150μmol photons $m^{-2}$ $s^{-1}$, 12L: 12D) in this study. In addition, the average growth rate could increase to 2.4 $d^{-1}$ under 12L: 12D light and dark cycle, when temperature rise to 25 °C (Zhang et al., 2020). In the present study, for the summer condition, daylength is 16 h, and temperature is 25 °C, which could lead to higher growth rate.

' 2.7 Measurements of PSII Proteins' orå '2.7 PSII Protein Measurements'å

(current header is technically correct but is archaic usage).

**Response:** Corrected.

Figures/Results; Figure 1: Amazing growth rate under summer conditions; the fastest I

have ever seen I think for a diatom.

**Response:** Please see our former responses (line 27-35).

Figure 2: Lovely data, congratulations. Very surprising switch of the OA effect in winter. Suggested additional plot: V at growth $pCO_2$? After all, most of the curve as plotted is above even the OA range of $pCO_2$. So a hypothetical Vmax may not be as important as the achieved V at growth conditions. Then, I would suggested plotting growth rate vs. V at growth condition ($pCO_2$ and light).

**Response:** We appreciate the reviewer for the useful suggestion. DIC concentrations in the culture media are about 2 and 2.1 mmol/L in ambient and OA environment respectively, and in the P-C curve, DIC concentration range from 0 to 4 mmol/L.

Photosynthetic rate increased significantly with increasing DIC concentration when

DIC is lower than 2 mmol/L, and the rate is relative constant when DIC is higher than

2 mmol/L. In the present study Vmax obtained from the P-C curve showed similar pattern and value with V at growth condition. We will add this information in the results part.

Figure 4: Without standard curves, be very cautious in interpretation of the immunoblotting data. The example blot result shows near-saturation (non-linear response of signal to target abundance) for many of the bands. So the Y axis dynamic range of the greyscale plots may be considerably compressed relative to the actual change in protein target abundance. Once a band is black, it cannot get any blacker. It is also very surprising that in winter OA increased RbcL signal.

**Response:** In this experiment we just calculated the relative value of each protein.

The data provide us with a general trend, not the accurate concentration as the reviewer mentioned, among different treatments. And we will add a caveat in the revised MS. Actin (internal control) could correct the experimental error in the process of quantitative sample loading of protein, to ensure the accuracy of the experimental results. The greyscale plots were measured according to density and the area of bands (TanonImage software). Although they are all black, the densities and areas are different in most conditions.

For the increased RbcL signal under OA condition in winter, although some researchers found RubisCO contents decreased at OA (Losh et al., 2013; Endo et al.,

2015), RbcL expression of different diatoms (*Thalassiosira pseudonana*,

*Phaeodactylum tricornutum*, and *T. weissflogii*) were found slightly increased under

OA in nitrogen-replete condition (Hong et al., 2017). And phytoplankton communities showed enhanced Rubisco expression in 800 ppm treatment compared with 350 ppm (Tortell et al., 2000). A coastal isolated *T. pseudonana* and *Emiliania huxleyi* also showed higher RbcL contents in higher $CO_2$ level, although the offshore isolated *T.*

*pseudonana* showed no significant difference between ambient and high $CO_2$

conditions (McCarthy et al., 2012). And low temperature could increase the relative amount of RbcL (Devos et al., 1998). We will add further discussion in the revised

MS.

**References:**

Balzano S, Sarno D, Kooistra W H C F. Effects of salinity on the growth rate and morphology of ten *Skeletonema* strains. Journal of Plankton Research, 2011, 33:

937-945.

Devos N, Ingouff M, Loppes R, et al. Rubisco adaptation to low temperatures: a comparative study in psychrophilic and mesophilic unicellular algae. Journal of

Phycology, 1998, 34: 655-660.

Endo H, Sugie K, Yoshimura T, et al. Effects of $CO_2$ and iron availability on rbcL

gene expression in Bering Sea diatoms. Biogeosciences, 2015, 12: 2247-2259.

Hong H, Li D, Lin W, et al. Nitrogen nutritional condition affects the response of energy metabolism in diatoms to elevated carbon dioxide. Marine Ecology Progress

Series, 2017, 567: 41-56.

Losh J L, Young J N, Morel F M M. Rubisco is a small fraction of total protein in marine phytoplankton. New Phytologist, 2013, 198: 52-58.

McCarthy A, Rogers S P, Duffy S J, et al. Elevated carbon dioxide differentially alters the photophysiology of *thalassiosira pseudonana* (bacillariophyceae) and *emiliania*

*huxleyi* (haptophyta) 1. Journal of Phycology, 2012, 48: 635-646.

Sakshaug E, Andresen K. Effect of light regime upon growth rate and chemical composition of a clone of *Skeletonema costatum* from the Trondheimsfjord,

Norway. Journal of Plankton Research, 1986, 8: 619-637.

Tortell P D, Rau G H, Morel F M M. Inorganic carbon acquisition in coastal Pacific phytoplankton communities. Limnology and Oceanography, 2000, 45: 1485-1500.

Zhang L, Li H, Wu M, Li F, and Xu J.: Effects of seawater acidification on photosynthetic physiological characteristics of *Skeletonema costatum* at different temperatures, Journal of Jiangsu Ocean University, 2020, 29: 1-7.

**Reviewer 2**

Comments:

The manuscript titled "Physiological responses of *Skeletonema costatum* to the interactions of seawater acidification and combination of photoperiod and temperature" described a research attempting to explore the impact of high $pCO_2$ (or ocean acidification, OA) under different seasons (combination of photoperiod and temperature) on the diatom *S. costatum*. The experiments are well conceived, and methods are clearly presented. The most interesting observation is that high $pCO_2$

(OA) does not uniformly impact *S. costatum* under different seasons: with somewhat negative impact on winter conditions. The authors showed the interesting observation, but the experimental design and data quality can be improved.

Here are some questions and suggestions for the authors.

**Response:** We thank the reviewer for the recognition of the value of our work and the valuable comments.

On the experimental setup, the authors stated that the cell culture pH did not change over 0.05 units in the 3d of one generation (section 2.3), so one very basic question is:

what are the pH values and ranges for the six different conditions (i.e. three seasons and two $pCO_2$)? Since the manuscript is about how OA impacts *S. costatum*

differently during different seasons, the acidification information, which can be presented easily as pH, is very critical to this whole article, however, this information is missing.

**Response:** The pH of culture media under different treatments are presented below.

And the information will be added in the revised MS.

| Treatments | W-LC | W-HC | A-LC | A-HC | S-LC | S-HC |
|---|---|---|---|---|---|---|
| pH | $8.10 \pm 0.01$ | $7.85 \pm 0.01$ | $8.14 \pm 0.01$ | $7.89 \pm 0.01$ | $8.19 \pm 0.02$ | $7.89 \pm 0.02$ |

Following the cell culture pH question, the authors measured the photosynthesis (P)

vs DIC curve at pH 8.12 (section 2.5) and very likely they did the same with P-I curve.

It would be better if the authors measure the responses under lower pH for high $pCO_2$

treated cells, according to the high $pCO_2$ (OA) conditions. It should be expected that the pH is lower under OA conditions, and *S. costatum* acclimated to OA conditions may not photosynthesize better under the experimental condition with higher pH

(8.12). As a result, the presented P-I and P-DIC curves for *S. costaum* cultured in HC

(OA conditions) may not reflect their real physiological status in terms of photosynthesis under OA conditions. Also please note that Tris buffer is known to change pH significantly with different temperature, so it is important to measure or calculate the pH at certain temperature.

**Response:** Researchers determine the P-C curves of OA treated cells under either ambient pH or culture pH (HC conditions). As the reviewer mentioned, P-C curves determined under culture pH could reflect the real physiological status under OA

conditions. However, the percentages of DIC species ($CO_2$, $HCO_3^-$, $CO_3^{2-}$) differ in media with different pH, which is inconvenient to compare $K_m$ of LC and HC cells.

Thus, some studies prefer setting same pH for LC and HC conditions to compare $V_{max}$

and $K_m$ at same pH (Nakajima et al., 2013; Shi et al., 2017). For P-I curves, the measurement time was much shorter than that used for P-C curves, so the pH

wouldn't change markedly and Tris buffer was not used for P-I curves in the present study. Cells were resuspended in pre-aerated fresh medium under cultured condition (i.e. pH 8.14 for LC cells, 7.9 for HC cells) to measure P-I curves. Details for P-I

measurements were added in the revised MS. We are sorry we missed the effects of

Tris buffer on pH at different temperature. For the P-C curves measurements in the present study, Tris was added in the culture medium, and the pH of the medium used for resuspending cells was adjusted at room temperature.

In the results session, the authors only mentioned and cited Fig. 1, while Figure 2, 3, 4

are listed at the end of the manuscript, none of those was referred in the text. In the tables presented, Table 2 and Table 3 do not show any units.

**Response:** We are sorry for the mistakes, Figure 2, 3, 4 are referred in the text now and units are added in Table 2 and 3 now.

In section 3.5 where the authors present the "PSII protein concentrations", RbcL was included as key PSII proteins. Such claim should be red-flagged, likewise the statement of "RbcL is related to the function of QA". Inclusion of RbcL in PSII

proteins is also found in the Abstract and Discussion. The authors should make sure what RbcL really does with creditable citations before writing assumptions or conclusions about RbcL.

**Response:** We apologize for the vague statement regarding the RbcL. It's revised in section 3.5, Abstract and Discussion.

The description of methodology "Values of Actin were divided by other densitometric scanning values of protein to calculate Gray-scale values" should be modified to indicate the supposing meaning of normalizing density to Actin.

**Response:** Corrected. It has been revised as "Actin was used as internal control in order to correct the experimental error in the process of quantitative sample loading of protein, to ensure the accuracy of the experimental results."

With the data presented in Figure 4, panel (a) and panel (b) do not seem to agree with each other. The western blot data does not look like a representative of the statistical data. For example, in W-HC (winter high $pCO_2$) condition, the D1 density is much higher than Actin (Fig 4a), so such value is greater than 2 if analyzed using ImageJ, however, the data presented in statistics showed a value very tightly close to 0.9 (Fig

4b). It would be nice if all raw data (immunoblots) are presented to support the statistics in Fig 4b.

**Response:** We thank the reviewer for pointing this out. We checked the data and revised the mistake. In Fig. 4 panel (a), the D1 value is 1.55 according to TanonImage software. The immunoblots data and revised Fig. 4 are presented as following:

[Figure]

[Figure]

Other minor concerns:

Section 2.7, "1000 and 34.5 are constants", what are the units? Or at least provide the unit of "where C represents total chlorophyll concentration".

**Response:** The unit of chlorophyll concentration is μg / ml. It is revised in the manuscript.

The use of letter "C" is ambiguous in the text. C was also used as carbon in line 219:

"C fixation".

**Response:** "C fixation" is revised as "carbon fixation".

For the measurement of specific growth rate, more details on how data were collected would be helpful. It would be better to have the raw data, cell concentration vs time (days), presented.

**Response:** More details will be added in the revised MS. Cells concentration was diluted every 3 days and cell concentration was counted two and three days after the dilution, such growth rate data in Fig. 1 were calculated for three times at different dilution days. The following figure presents cell concentrations and specific growth rate at three different times.

[Figure]

Line 229, "initial slop" should be "initial slope".

**Response:** Corrected.

References

Shi Q, Xiahou W, Wu H. Photosynthetic responses of the marine diatom *Thalassiosira*

*pseudonana* to $CO_2$-induced seawater acidification. Hydrobiologia, 2017, 788:

361-369.

Nakajima K, Tanaka A, Matsuda Y. SLC4 family transporters in a marine diatom directly pump bicarbonate from seawater. Proceedings of the National Academy of

Sciences, 2013, 110: 1767-1772.

---

## Author Comment (AC2) · 26 Oct 2020

Dear reviewer,

Thank you for the useful comments and suggestions that have greatly improved this manuscript. Attached you can find our reply to your comments.

Best regards, Hangxiao Li

Please also note the supplement to this comment:

https://bg.copernicus.org/preprints/bg-2020-303/bg-2020-303-AC2-supplement.pdf

---

## Author Response (AR2)

Dear editor and reviewer:

We would like to thank you for your valuable comments that have greatly improved this manuscript. We have revised the manuscript with responses to the comments point by point.

The main change made in the revised version is that the P-C curve measurement part is deleted as the result is not convincing due to an inappropriate manipulation on the Tris buffer at different temperature and the main conclusions of the present study wouldn't be affected without it.

Correspondence about this paper should be directed to Futian Li and Juntian Xu at the following address, phone and fax number and email address.
Sincerely yours,
Futian Li and Juntian Xu
Jiangsu Key Laboratory of Marine Bioresources and Environment, Jiangsu Ocean University, Lianyungang 222005, China
Tel: +86 518 85895421;
Fax: +86 518 85895427;
E-mail: futianli@jou.edu.cn (Futian Li) and jtxu@jou.edu.cn (Juntian Xu)

**Comments**

The authors showed the negative effect of OA in winter to *S. costatum* by presenting the slower growth rate and stating lower photosynthetic rate under HC (OA condition). However, the P-C curve measurement was done with Tris-buffered media at different temperatures (pH tuned to 8.12 at room temperature). A Tris buffer pH 8.12 at room temperature means the pH at 5 °C to be around 8.6, which would cause $pCO_2$ much lower in the Tris-buffered media than that in the culture media with the same total DIC. Researchers should be critical when interpreting such results.

**Response:** We apologize for the inappropriate manipulation on the Tris buffer at different temperature. As the referee said, the pH of Tris buffer changed significantly with temperature variation. Thus, the percentage of different DIC species varied among different temperature treatments even with same total DIC, which would mislead readers. As the DIC utilization of *S. costatum* was not one of the main results in the present study, we will delete this part in the revised MS.

On the net photosynthetic rate measurements, the P-I curves of winter conditions did not show difference between HC

and LC (Figure 4), especially the data points falling in the range of 100 to 200 µmol photons m$^{-2}$ s$^{-1}$. This does not support a lower photosynthetic rate and its explanation for slower growth rate for *S. costatum* under winter OA condition (cultured under 150 µmol photons m$^{-2}$ s$^{-1}$).

**Response:** The P-I curves were measured under 10, 20, 50, 100, 200, 500, 1000 µmol photons m$^{-2}$ s$^{-1}$ without the light intensity under culture condition (150 µmol photons m$^{-2}$ s$^{-1}$). Although the P-I curves of winter conditions did not show significant difference between HC and LC (Figure 4), the trend was in line with lower photosynthetic rate for *S. costatum* under OA condition in winter (cultured at 150 µmol photons m$^{-2}$ s$^{-1}$).

As for the quantification of RbcL and PSII proteins, the elevated expression levels of photosynthetic proteins and a lower rate of photosynthesis (as the authors stated) under winter OA condition is confounding, and the authors did not provide sufficient convincing explanation.

**Response:** The elevated expression levels of photosynthetic proteins may be inconsistent with photosynthetic rate because that protein contents we measured include both photochemically active PSII, and those PSII which are inactivated but retain D1 and D2 subunits (Li et al., 2015). And the removal rate of photosynthetic proteins from photoinactivated PSII complexes could be different according to culture condition. For example, the removal rate of D1 protein increased with growth light in the diatom *Thalassiosira pseudonana* (Campbell et al., 2013). We have added further discussion on it in the revised MS.

Technical corrections suggested from Editor:

L20: low temperature and short daylength on the abundace of RbcL and key photosystem II (PSII) proteins (D1 and D2).

**Response:** corrected

L97: Remove the underscore between costatum and was.

**Response:** corrected

L102: Not milli-Q, but Milli-Q.

**Response:** corrected

L110: insert a space between s^-1 and light.

**Response:** corrected

L141: The abundance of Riblose-1,5-biphosphate carboxylase/oxygenase...

**Response:** corrected

L341: rbcL should be italic.

**Response:** corrected

**References**

[revised manuscript text omitted]